# Unveiling the developmental dynamics and functional role of Odorant Receptor Co-receptor (*Orco*) in *Aedes albopictus*: A novel mechanism for regulating odorant receptor expression

Hui Yao[1,2‡], Qian Qi[1‡], Dan Gou[3], Simin Liang[1], Stephen T. Ferguson[1], Ming Li[2], Heng Zhang[1], Zi Ye[1,4]*, Feng Liu [1,5]*

1 Institute of Infectious Diseases, Shenzhen Bay Laboratory, Shenzhen, China, 2 Department of Parasitology, Zhongshan School of Medicine, Sun Yat-sen University, Guangzhou, China, 3 Department of Cell Biology, Zunyi Medical University, Zunyi, China, 4 School of Life Sciences, Guizhou Normal University, Guiyang, China, 5 Guangdong Provincial Key Laboratory of Infection Immunity and Inflammation, Shenzhen, China

‡ HY and QQ share the first author for this article.
* liufeng@szbl.ac.cn (FL); zi.ye@gznu.edu.cn (ZY)

## Abstract

As one of the most aggressive disease vectors, the Asian tiger mosquito *Aedes albopictus* relies heavily on its olfactory system to search for food in the larval stage, locate hosts after eclosion, and identify suitable oviposition sites after blood feeding. In mosquitoes and other insects, the olfactory system detects environmental odors primarily through a diverse repertoire of odorant receptors (ORs), which require the highly conserved odorant receptor co-receptor (*Orco*) to function. While *Orco*'s role in enabling receptor function is well established, its cellular localization patterns, developmental expression dynamics, and system-wide impact on olfactory physiology and behavior remain understudied in *Ae. albopictus*. To address this knowledge gap, we leveraged the Q-system to systematically characterize *Orco*-expressing neurons across embryonic, larval, and adult stages of *Ae. albopictus*. *Orco*-expressing neurons were observed as early as in the embryonic stage and proliferated during larval development. *Orco* expression in adults spanned the olfactory sensory neurons (OSNs) of the antennae, labella, and maxillary palps in both male and female mosquitoes, consistent with its conserved peripheral distribution across various mosquito species. To further investigate the functional implications of *Orco*, we generated *Orco* knockout mutants and strikingly discovered that *Orco* knockout mutants displayed significant widespread downregulation of ORs, suggesting that *Orco* may influence ORs' expression or stability. Electrophysiological recordings confirmed significantly attenuated responses to human volatiles in *Orco* mutants, and behavioral assays demonstrated a marked decline in blood-feeding efficiency and decrease of human preference in females. Together, these findings reveal dynamic organization of OSNs

**Data availability statement:** All relevant data are within the manuscript and its supporting information files.

**Funding:** This work was supported by the National Natural Science Foundation of China under Grants 82372289 and 82350410493 awarded to F.L. and S.T.F., respectively, and by Guizhou Education Department (Grant No. Qian Jiao Ji [2024] 54) and Guizhou Normal University (Grant No. QSXM [2022] B09) awarded to Z.Y., and the China Postdoctoral Science Foundation (2024M752152) to Q.Q. The funders of this study didn't play any role in the study design, data collection and analysis, decision to publish, or preparation of the manuscript.

**Competing interests:** The authors have declared that no competing interests exists.

during mosquito development and uncover the critical role of *Orco* in maintaining the integrity and function of the olfactory system*, providing insights which may inform novel, next-generation vector control strategies.*

## Author summary

The mosquito olfactory system detects environmental odors primarily through a diverse repertoire of odorant receptors (ORs), which require the highly conserved odorant receptor co-receptor (*Orco*) to function. While *Orco*'s role in enabling ORs' function is well established, its cellular localization patterns, developmental dynamics, and system-wide impact on olfactory physiology and behavior remain understudied in *Ae. albopictus*. To address this knowledge gap, we leveraged the Q-system to systematically characterize *Orco*-expressing neurons across embryonic, larval, and adult stages of *Ae. albopictus*. We observed *Orco*-expressing neurons as early as in the embryonic stage and proliferated during larval development and found that *Orco* expression in adults spanned the olfactory sensory neurons (OSNs) of antennae, labella, and maxillary palps. To further investigate the functional implications of *Orco*, we generated *Orco* knockout mutants and strikingly discovered that *Orco* mutants displayed significant widespread down-regulation of ORs, suggesting *Orco* may influence OR expression or stability. Moreover, *Orco* mutants displayed significantly attenuated electrophysiological responses to human volatiles, marked decline in blood-feeding efficiency, and the preference of female mosquitoes for human hosts was significantly decreased. Together, these findings reveal dynamic organization of OSNs during mosquito development and uncover the critical role of *Orco* in maintaining the integrity and function of the olfactory system.

## Introduction

The Asian tiger mosquito, *Aedes albopictus*, is a major vector of human disease, including dengue, chikungunya, Zika, and lymphatic filariasis, which poses significant risks to global public health [1,2]. Dengue fever is one of the most severe tropical diseases transmitted by *Aedes* mosquitoes with models placing estimates around 390 million cases every year [3] and 4 billion people at risk [4]. Importantly, *Ae. albopictus* has replaced *Ae. aegypti* as the primary vector of dengue virus in several countries, such as China and Japan, with its invasive range expanding at a far greater rate than *Ae. aegypti* due to its superior ability to compete for breeding sites [5,6]. In light of the growing public health threat posed by *Ae. albopictus* and its rapidly expanding geographic range, a comprehensive understanding of the olfactory system, an essential driver of host-seeking behavior, is critical for developing timely and effective vector management strategies.

*Ae. albopictus* rely on their olfactory system for essential behaviors such as host-seeking and oviposition. The primary olfactory appendages in mosquitoes include the antennae, maxillary palps, and labella. Along these appendages are small sensory hairs known as sensilla that house sensory neurons responsible for the detection of various chemical, thermal, and mechanical stimuli [7]. Within the chemosensory neurons, insects express one or more major classes of chemoreceptor: odorant receptors (ORs), ionotropic receptors (IRs), and gustatory receptors (GRs) [8–11]. Chief among these are the olfactory sensory neurons (OSNs) which are responsible for detecting volatile odorants and pheromones, such as those associated with human sweat, and guiding mosquitoes towards potential hosts [12,13]. Each of these neurons express one of many potential ORs which convey ligand-specificity, alongside a highly conserved obligate co-receptor (*Orco*) [14]. Together, these protein subunits form a heterotetrameric (1:3 OR:Orco) ligand-gated ion channel on the membrane of the OSN [15–18]. Upon ligand binding, the channel undergoes a conformation change that leads to the opening of the central pore, allowing the influx of ions that depolarize the neuron and generate an action potential [17]. This electrical signal is transmitted to the brain, where it is processed to guide behaviors such as host seeking and oviposition.

Among the various molecular components involved in ORs-mediated olfactory signal transduction, *Orco* is of central importance, and its conservation across diverse insect species highlights its essential contribution to the evolution and function of the insect olfactory system. Indeed, only certain species from the most basal insect order, *Archaeognatha*, appears to lack this essential gene [19]. Moreover, numerous studies across diverse insect lineages, including mosquitoes, have consistently demonstrated that the loss of *Orco* function dramatically impairs olfactory sensitivity and physiology, leading to aberrant behaviors. In *Drosophila*, *Orco* knockout significantly impairs olfactory responses to a broad range of general odorants and disrupts behavior in valence bioassays in larvae and adults [20]. In ants, both genetic knockout of *Orco* and pharmacological modulation of *Orco* protein significantly alter olfactory sensitivity to hydrocarbons and other important social chemical cues leading to the loss of social behaviors [21–23]. In mosquitoes, *Orco* knockout impairs the mosquitoes' sensitivity to both human- and non-human-derived odor cues resulting in diminished attraction to human hosts and oviposition sites [24,25]. Taken together, these studies reinforce *Orco*'s central role in olfactory signaling and make it a prime target for developing vector control strategies.

In this study, we hypothesize that *Orco* plays a fundamental role in the development patterning of its peripheral olfactory system and the olfactory mechanisms underlying its host-seeking and blood-feeding behavior. To address this hypothesis, we performed a comprehensive characterization of *Orco*-expressing neurons across the developmental stages of *Ae. albopictus* mosquitoes. To elucidate the spatiotemporal expression patterns and function of *Orco*-expressing sensory neurons, we employed a combination of advanced genetic and electrophysiological techniques. Specifically, we used the Q system [26] to label *Orco*-expressing neurons via GFP, and integrated homology assisted CRISPR/Cas9 knockin (HACK) and PiggyBac transposon-mediated genomic integration to generate the AalbOrco-QF2 driver line and AalbQUAS-mCD8:GFP effector line. In addition, we used RNA-sequencing, electroantennography (EAG), and single sensillum recording (SSR) to assess the impact of *Orco* homozygous knockout, while behavioral assays were used to examine the effects on blood-feeding efficiency and host preference. This study demonstrated a novel role of *Orco* in regulating ORs transcription in mosquitoes and uncovered the dynamic organization of olfactory sensory neurons during *Ae. albopictus* development. These findings demonstrated the critical role of *Orco* in chemical perception and host-seeking behavior in *Ae. albopictus*, providing a theoretical foundation for developing novel olfaction-based mosquito control strategies.

## Materials and methods

### Mosquito maintenance

Mosquito *Ae. albopictus* Foshan strain was a gift from Dr. Xiaoguang Chen at Southern Medical University, Guangzhou, China and reared as described [27]; and 5- to 7-day-old non-blood-fed females were used for all experiments. All

mosquito lines were reared in environmental chambers at 27°C and 75% relative humidity under a 12:12 light-dark cycle and supplied with 10% sucrose water in the Shenzhen Bay Laboratory Insectary.

## sgRNA design and production

The procedure for single guide RNA (sgRNA) synthesis followed previously described methods [28] with minor modifications. sgRNA were designed for high efficiency by searching the sense and antisense strands of the *Orco* gene (AALFPA_042885) for the presence of protospacer-adjacent motifs (PAMs) with the sequence of NGG using CHOP-CHOP [29]. sgRNAs were synthesized using the EnGen sgRNA Synthesis Kit (New England Biolabs) according to the manufacturer's protocol using 300 ng of purified DNA template. Following in vitro transcription, the sgRNAs were purified using the Monarch RNA Cleanup Kit (New England Biolabs) and diluted to 1000 ng/µl in nuclease-free water and stored in aliquots at −80 °C. Recombinant Cas9 protein from *Streptococcus pyogenes* was obtained commercially (TrueCut Cas9 Protein V2, Invitrogen by Thermo Fisher Scientific) and diluted to 1000 ng/ul in nuclease-free water and stored in aliquots at −80 °C.

## CRISPR mediated microinjections

To generate the *AalbOrco-QF2* driver line for the Q system, *T2A-QF2-3xP3-DsRed* element was inserted into the *Orco* coding region through CRISPR-mediated homologous recombination. The homologous template (donor plasmid) that contains *T2A-QF2-3xP3-DsRed* element, which was amplified from the ppk301-T2A-QF2 HDR plasmid (Addgene plasmid# 130667) [30], flanked by ~1kb homologous arms was constructed using NEBuilder HiFi DNA Assembly kit (New England Biolabs). The left homologous arm was amplified with primer pair of OrcoleftarmFwd(GACGGCCAGTCAGGG GCGCTTCAAGTTAATAATTAAAAAAATAC) and OrcoleftarmRev(CTCTGCCCTCCGGACGGTAGGTGTCCAG). The right homologous arm was amplified with primer pair of OrcorightarmFwd (ATGTATCTTAACTCGGCTGCCCTGTTCC) and Orc-orightarmRev (CAGCTATGACCGGCTCCGTGTGTAAGATCAC). The primer pair for comfirming the knockin element was AalbOrco-F (GCCGACGTGATGTTCTGCTCCTGGTTGCTG) and AalbOrco-R(ACTTGCACACACCACCACCATAGG GACACG). From the left homologous arm immediately preceding the T2A, 1 bp was removed to keep theT2A sequence in-frame. Red-eyed F1 mosquitoes were backcrossed for five generations and then crossed to the effector line to acquire progeny for *Orco* localization studies.

Embryonic collection and CRISPR microinjections were performed following the procedure described in Li et al [28]. Briefly, Aedes mosquitoes were blood-fed 4 days before egg collection. An ovicup filled with ddH$_2$O and lined with filter paper was placed into a cage and female mosquitoes were allowed to lay eggs in the ovicup in the dark. After 30–60 min, the ovicup was taken out and unmelanized eggs were transferred onto a glass slide. The eggs were quickly aligned on a wet piece of filter paper. Aluminosilicate needles were pulled on a Sutter P-1000 needle puller and beveled using a KDG-03 beveler (Kewei, Wuhan). An Eppendorf Femtojet 4i was used for power injections under a compound microscope at 10 × magnification (E5, Soptop, Ningbo). About 10 eggs were injected each time immediately after fresh eggs were collected. The concentration of components used in the study was as follows; Cas9 protein at 300 ng/µl, sgRNA at 40 ng/µl, donor plasmid 300 ng/µl. After injection, eggs were placed in a cup filled with water and allowed to hatch and develop into adults. The first generation (G0) of injected adults were separated based on sex and crossed to 5X wild-type counterparts. Their offspring (F1) were manually screened for DsRed-derived red eye fluorescence using an Compound Fluorescent Microscope (DP74, Olympus, Japan). Red-eyed F1 males were individually backcrossed to 5- fold females to establish a stable mutant line. DNA extraction was performed using FastPure Gel DNA Extraction (Vazyme Biotech, Nanjing) protocols and genomic DNA templates for PCR analyses of all individuals were performed (after mating) to validate the fluorescence marker insertion using primers that cover DSB sites. PCR products were sequenced to confirm the accuracy of the genomic insertion. Heterozygous mutant lines were thereafter backcrossed to wild-type *Ae. albopictus* for at least five generations before self-crossing and the progenies were used for screening homozygous individuals according to their

DsRed-derived red eye fluorescence intensity. Putative homozygous mutant individuals were mated to each other before being sacrificed for genomic DNA extraction and PCR analyses (as above) to confirm their homozygosity.

The effector line (QUAS-mCD8:GFP) was generated by the plasmid pBAC-ECFP-15xQUAS_TATA-mCD8-GFP-sv40 (Addgene #104878) [31] with a pBac helper plasmid. Similar to establishing the driver line, the first generation (G0) of injected adults were separated based on sex and crossed to 5-fold wild-type counterparts. Their offspring (F1) were manually screened for ECFP-derived cyan eye fluorescence. Single cyan-eyed F1 male was backcrossed for five generations to five wildt-ype females in order to establish a stable transgenic effector line which later would be used for crossing with the driver line.

## Whole-Mount Imaging

Offspring resulting from crosses between the homozygous *AalbOrco-QF2* driver line and the heterozygous *AalbQUAS-mCD8:GFP* effector line were screened for eye-specific expression of DsRed and ECFP. The anatomical method of mosquito eggs referred to Juhn and James (2012) [32]. The methods for fixing the sample and capturing confocal images referred to Ye et al (2022) [33]. Embryo, larval antennae as well as antennae, maxillary palps and proboscis from 4- to 6-d-old adults were dissected into 4% formaldehyde in PBST (0.1% Triton X-100 in phosphate-buffered saline) and fixed on ice for 30 min. Samples were thereafter washed 3X in PBST for 10 min each and directly transferred onto slides and mounted in Anti fluorescence quenching sealing solution (Beyotime). Confocal microscopy images at 1024×1024 pixel resolution were collected with the LSM980 system (Zeiss). Laser wavelength of 488 nm was used to detect green fluorescent protein (GFP).

## Electrophysiology

Single sensillum recordings (SSR) were conducted as described in Liu et al [34]. Mutant and wild-type female mosquitoes 4 days after eclosion were anaesthetized on ice for 2–3 min and mounted on a microscope slide (76×26 mm). The antennae were fixed using a double-sided tape to a cover slip resting on a small ball of dental wax to facilitate manipulation. Once mounted, the specimen was placed under a microscope (Eclipse FN1, Japan) and the antenna viewed at a high magnification (1000×). Two tungsten microelectrodes were sharpened in 10% $KNO_2$ at 10 V. The reference electrode, which was connected to ground, was inserted into the compound eye of the mosquito and the other was connected to the preamplifier (10×, Syntech, Kirchzarten, Germany) and inserted into the shaft of an olfactory sensillum to complete the electrical circuit to extracellularly record ORN potentials [35]. Controlled manipulation of the electrodes was performed using a micromanipulator (Burleigh PCS-6000, CA). The preamplifier was connected to an analog-to-digital signal converter (IDAC-4, Syntech, Germany), which in turn was connected to a computer for signal recording and visualization in the software AutoSpike v5.1. Signals were recorded for 10 s starting 1 s before stimulation, and the action potentials were counted offline automatically with the AutoSpike software over a 500-ms period before and after stimulation. The spontaneous firing rates observed in the preceding 500 ms were subtracted from the total spike rates observed during the 500-ms stimulation, and counts were calculated in units of spikes/s.

Thirty-four compounds from different chemical classes were selected for electrophysiological recording of olfactory sensilla with various morphological shapes. Each compound was prepared in 100-fold dilution (v/v) with dimethyl sulfoxide (DMSO) or paraffin oil. For each dilution, a 10 µl portion was dispersed onto a filter paper strip (4×30 mm), which was then inserted into a Pasteur pipette to create the stimulus cartridge. A sample containing the solvent alone served as control. The airflow across the antennae was maintained constant at 20 ml/s throughout the experiment. Purified and humidified air was delivered to the preparation through a glass tube (10-mm inner diameter) perforated by a small hole 10 cm away from the end of the tube, into which the tip of the Pasteur pipette could be inserted. The stimulus was delivered to the sensilla by inserting the tip of the stimulus cartridge into this hole and diverting a portion of the air stream (0.5 l/min) to flow through the stimulus cartridge for 500 ms using a stimulus controller (Syntech, Germany). The distance between

the end of the glass tube and the antennae was ≤ 1 cm. The number of spikes/s was obtained by averaging the results for each sensillum/compound combination.

The electroantennogram procedure followed previously described protocols [36] with minor modifications. Briefly, the head of an adult *Ae. albopictus* female was excised and mounted on an EAG platform equipped with two micromanipulators and a high-impedance AC/DC preamplifier (Syntech, Germany). Chlorinated silver wires in glass capillaries filled with 0.1% KCl and 0.5% polyvinylpyrrolidone were used for both reference and recording electrodes. One antenna with the tip cut was accommodated into the recording electrode. The airflow across the preparation was maintained constant at 20 ml/s to which a stimulus pulse of 2 ml/s was delivered for 500 ms. Any change in antennal deflection induced by the stimuli or control puffs was recorded for 10 s. All compounds were dissolved in DMSO or paraffin oil to make a test solution of 10-fold dilution. An aliquot (10 µl) of a tested compound was loaded onto a filter paper strip (4 × 30 mm), which was immediately inserted into a Pasteur pipette for evaporation. Solvent (paraffin oil) alone served as control. For each compound, EAG responses of 5–11 female mosquitoes were recorded. These recordings were analyzed using EAG software (EAG Version 2.7, Syntech). The EAG response (ΔmV) to each test stimulus was calculated by subtracting the value of the antennal response elicited by the solvent from that stimulated by the compound.

### Chemicals

Compounds that were used in electrophysiological (EAG and SSR) recordings are listed in S1 Table.

### Transcriptome analysis

Total RNA was extracted from dissected antennal samples from 4 to 7-day-old post-eclosion wild-type and *AalbOrco^{DsRed/DsRed}* mosquitoes (three biological replicates each, with each replicate containing 300 female mosquitoes' antennae) by TRIZOL (Invitrogen, Carlsbad, CA, USA) according to manual instruction. mRNA was purified from total RNA using magnetic beads coated with Oligo (dT), purified mRNA was fragmented into small pieces with fragment buffer at appropriate temperature. These mRNA were used as templates to synthesize the first strand of cDNA with random hexamers. The second strand cDNA was synthesized by adding reaction buffer, dNTPs, DNA polymerase I and RNase H, and then was purified by AMPure XP beans. After end repair, A tail and ligation of the sequencing connector, screens were performed with AMPure XP beads. The final library was obtained after PCR amplification and purification. The final library was amplified with phi29 to make DNA nanoball (DNB) which had more than 300 copies of one molecular, DNBs were loaded into the patterned nanoarray and pair end 150 bases reads were generated on DNBSEQ-T7 platform at Tsingke Biotechnology Co., Ltd. Six transcriptomic libraries (three replicates for both wild-type and mutant mosquitoes) were made in this study. Based on the assembly results, Bowtie2 was used to map the clean reads of each sample to Unigene, and then RSEM was used to calculate the gene expression level of each sample. Blastn, Blastx, and Diamond were used to align Unigenes to NT (https://www.ncbi.nlm.nih.gov/nucleotide/), NR (https://www.ncbi.nlm.nih.gov/refseq/about/nonredundant-proteins/), KOG (https://www.hsls.pitt.edu/obrc/index.php?page=URL1144075392), KEGG (https://www.genome.jp/kegg/), and Swiss-Prot database for annotation. All the genes annotated as odorant receptors, ionotropic receptors, gustatory receptors and their co-receptors were search out and analyzed by Prism 5 (GraphPad Software). Their relative expression levels were determined by FPKM (fragments per kilobase of transcript per million fragments mapped) counts.

### Blood-feeding assay

Blood-feeding and host preference (below) assays were carried out as described in Wang et al [37] with minor modifications. The experimental subjects were wild-type and *AalbOrco^{DsRed/DsRed}* homozygous mutant *Ae. albopictus*. Each trial used 30–35 non-blood-fed, 3-day-old, mated adult females. These mosquitoes were starved for 12 hours prior to the experiment. A 20-minute blood-feeding assay was conducted using mice immobilized with adhesive tape and a funnel-shaped wire mesh cage, during which mosquitoes primarily fed on the mouse's tail. The experiments were performed in 18 × 18 × 18 cm nylon

mesh cages, with 3 replicates conducted for each mosquito strain. The number of blood-fed mosquitoes was determined by visual inspection as fed or unfed. The ratio of blood-fed mosquitoes was calculated using the following formula: blood-fed (%) = Nb/Nt, where Nb was the number of blood-fed mosquitoes and Nt was the total number of mosquitoes.

### Host preference assay

The host preference assay was performed on wild-type and $AalbOrco^{DsRed/DsRed}$ homozygous mutant *Ae. albopictus*. Each trial used 30–35 non-blood-fed, 3-day-old, mated adult females, which were fasted for 24 h prior to assay. Before testing, the mosquitoes were transferred to a two-choice olfactometer (20 x 20 x 32 cm) and allowed to acclimate for 30 min. The two ports of the olfactometer were respectively equipped with a human hand and a mouse. The participating volunteer (a 27-year-old female) had refrained from using any scented products for 48 hours prior to the experiment and had not washed her hands for 4 hours. The test mouse was immobilized adhesive tape and a funnel-shaped wire mesh cage. Fans were used to direct odors from the human hand and mouse, respectively, into the olfactometer toward female mosquitoes at an airflow speed of 0.5 m/s. The experiment lasted for 10 minutes, after which the number of female mosquitoes captured in the trap was counted and recorded. Host preference was expressed as the preference index = (Nh – Nm)/(Nh + Nm), where Nh was the number of mosquitoes probing human hand and Nm the number of mosquitoes probing mice. Six replications for each assay were performed.

### Mosquito fecundity assay

Mated Female mosquitoes 3 days post-eclosion were blood-fed on mice. Two days after the blood meal, individual females were transferred into Drosophila culture tubes (1.2 cm in radius × 9.5 cm in height), each containing a filter paper strip (1 cm × 5 cm) moisturized with 2 mL of ddH$_2$O. Oviposition was allowed for four days, after which the number of eggs laid was counted. Following the egg count, the oviposition paper was returned to its original tube and submerged in ddH$_2$O. The number of hatched larvae was recorded daily. For both wild-type and $AalbOrco^{DsRed/DsRed}$ lines, egg production and hatching rate were assessed for eight individual females per line.

## Results

### Targeted CRISPR knockin of *AalbOrco*

To first characterize the spatial expression patterns of *Orco* in *Ae. albopictus* (*AalbOrco*), we employed the binary expression Q system which enables precise visualization of gene expression through fluorescent signal reporting [26] and has been successfully implemented in *Anopheles gambiae*, *Ae. aegypti* and *An. coluzzii* [30,31,33]. This system crosses an *AalbOrco* promoter-QF2 (*AalbOrco-QF2*) driver line with a QUAS-GFP effector line to promote GFP expression in *AalbOrco*-expressing cells. Here, we generated the driver line using the homology-assisted CRISPR/Cas9 knock-in (HACK) method [38], targeting the fourth coding exon of *AalbOrco* by using guide RNA to insert an in-frame *T2A-QF2-3xP3-DsRed* cassette (Fig 1A, B). Successful integration was confirmed by genomic PCR amplification of the target locus (Fig 1C). Concurrently, we established the QUAS-GFP effector line using the piggyBac transposon system. This effector line carries the QUAS-mCD8:GFP-*3xP3-ECFP* effector cassette inserted into any TTAA site of the genome (Fig 1D). Virgin females of these two lines were then backcrossed for five generations to wild-type males to minimize genetic variability and establish stable transgenic mosquito lines, respectively. And we utilized eye-specific DsRed and ECFP fluorescent markers to screen transgenic larvae, respectively (Fig 1E, F).

### Expression of *AalbOrco* in the late embryo of *Ae. albopictus*

Mosquito embryogenesis undergoes three major developmental stages: the initial stage, characterized by limited bristle development and cellular division during germ band extension; the intermediate stage, marked by bristle formation,

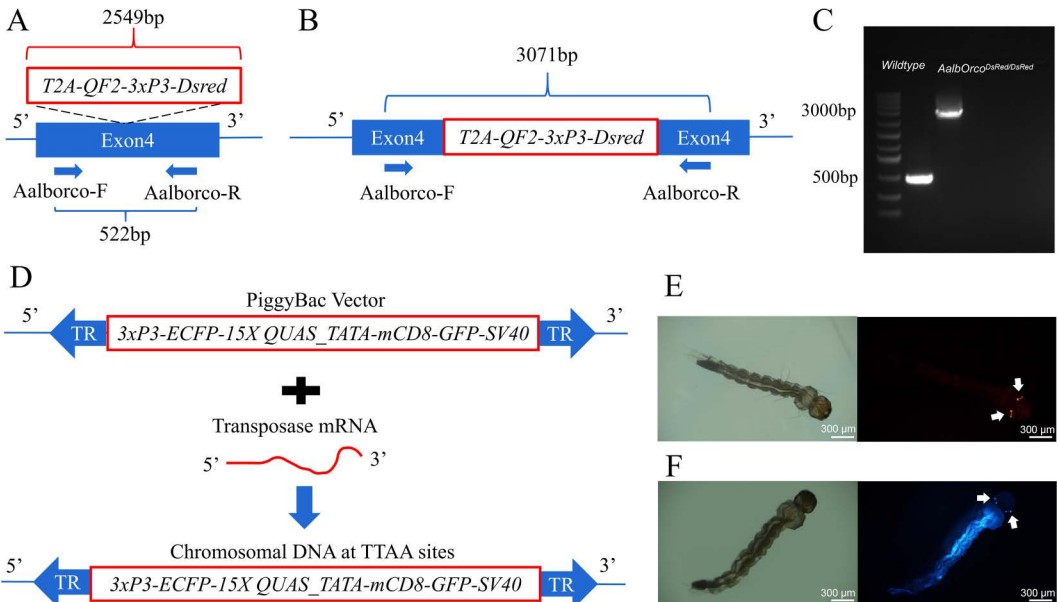

**Fig 1. CRISPR-mediated construction of *AalbOrco-QF2* driver and AalbQUAS-mCD8:GFP effector lines.** Homology-assisted CRISPR knock-in strategy. The *T2A-QF2-3xP3-DsRed* element (2549-bp) was inserted into the fourth exon of *AalbOrco* via CRISPR-mediated homologous recombination. PCR validation utilized primers AalbOrco-F/AalbOrco-R and produced a 552-bp wild-type amplicon and a 3,071-bp mutant-specific product in homozygous individuals. **(B)** Primers AalbOrco-F and AalbOrco-R were designed to amplify the region outside the homology arm. This ensured allele-specific amplification, as the wild-type genome produced only the 552 bp fragment, while the knock-in allele generated the 3,071 bp band. **(C)** Agarose gel electrophoresis successfully confirmed knock-in integration in transgenic mosquitoes. The 3,071-bp band exclusively appeared in homozygous *AalbOrco^{DsRed/DsRed}* individuals, while wild-type controls showed only the 552-bp fragment. **(D)** The *15xQUAS-GFP* cassette (5,447-bp) was inserted into random TTAA sites in the genome via PiggyBac system. **(E)** In *AalbOrco-QF2* driver lines, *DsRed* fluorescence localized specifically to the eyes (scale bar: 300 μm), indicating functional promoter activity. **(F)** *AalbQUAS-mCD8:GFP* effector lines exhibited ECFP fluorescence in the eyes, demonstrating transposon-mediated cassette insertion (scale bar: 300 μm)..

segmentation of the cephalic and thoracic regions, and the development of structures such as the respiratory siphon; and the final stage, where complete segmentation occurs and the chorion-breaking spike forms preparing for larval hatching [39]. While the mosquito egg remains understudies relative to other developmental stages, previous studies have demonstrated that certain chemical cues, such as those found in yeast, trigger embryo hatching, suggesting a potential chemosensory capacity of mosquito embryos [40]. To investigate the expression pattern of *AalbOrco* in the late embryo stage of *Ae. albopictus*, we dissected the embryo chorion with an insect pin and directly imaged the GFP-labeled sensory neurons in the cephalic region of the embryo. While no GFP-labelled tissues had been observed in the initial (12 hours post-egg laying) and intermediate (36 hours post-egg laying) stage of mosquito embryoes, we found *Orco^+* neurons in the cephalic section of late embryos (72 hours post-egg laying) in two prepared samples after multiple unsuccessful trials (Fig 2). Interestingly, one sample only display single *Orco^+* neuron (Fig 2A) while the other sample showed 3–4 GFP-labeled neurons (Fig 2B) in the antenna-like tissue, suggesting the heterogeneity of *Orco* exprepression during the development of olfactory sensory neuron. This result demonstrated the initial expression of *Orco* in the late embryonic stage of *Ae. albopictus*, providing support for the potential olfactory capacity in early development.

### Expression pattern of *AalbOrco* in the larval antennae

Mosquito larvae possess a robust chemosensory capacity through a relatively simple olfactory system. To determine the spatial distribution of *AalbOrco* in the antennae of mosquito larvae, we systematically examined the progeny from crossing

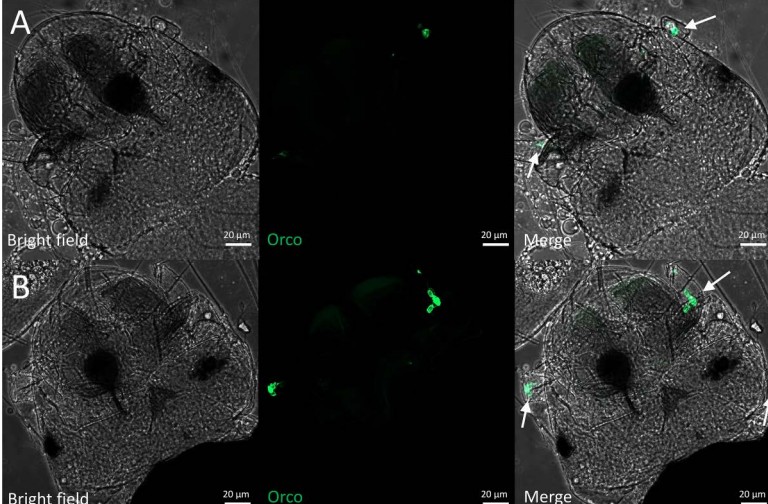

**Fig 2. Localization of *Orco*-expressing neurons in the late embryo of *Ae. albopictus*. (A) & (B)** Representative *Orco*-expressing neurons in the antenna-like tissue of the embryo at sample 1 (A) and sample 2 (B) 72 hours post-egg laying.

the *AalbOrco-QF2* driver line (*AalbOrco*$^{+/DsRed}$) with *AalbQUAS-mCD8:GFP* effector line. In these hybrids, GFP fluorescence specifically labeled neuronal dendrites, cell bodies, and axons in the larval sensory cone across all four instars (Fig 3A-D). High-resolution confocal imaging showed that the number of GFP-labelled ORNs increased during larval development ($N_{L1} = 5.25 \pm 1.40$ (n = 8), $N_{L2} = 7 \pm 1.10$ (n = 6), $N_{L3} = 11 \pm 0.89$ (n = 6), $N_{L4} = 11.93 \pm 1.49$ (n = 14); Fig 3E). Notably, the number of *Orco*-expressing neurons increased the most during the transition from 2$^{nd}$ to 3$^{rd}$ instar larva, which correlates with the considerable feeding needs and highly active food-searching behavior of 3$^{rd}$ instar larva.

### Expression pattern of *AalbOrco* in adult mosquito *Ae. albopictus*

To systematically investigate the spatial and sex-specific expression pattern of *AalbOrco* in *Ae. albopictus* adults, progeny derived from crosses between parental driver and effector lines were used for the whole-mount examination. Transgenically driven GFP expression was robust in OSNs of antennae, labella and maxillary palps. Expression patterns in the adult antennae were sexually dimorphic: female mosquitoes exhibited strong labeling across all 13 antennal segments, while male individuals displayed restricted expression on only the distal two segments (Fig 4A, B). This sexually dimorphic pattern aligns with similar findings in *An. gambiae* [31]. *AalbOrco* expression in the labella of female (Fig 4C) was exclusively localized to OSNs with short dendrites associated with olfactory T2 sensilla as opposed to gustatory T1 sensilla, consistent with reports in *Ae. aegypti* [41] and *An. coluzzii* [42]. Despite pronounced morphological divergence in maxillary palp segmentation between female and male mosquitoes (males have four segments (Fig 4E) while females only have three (Fig 4D)), the GFP-labelled *Orco*-expressing neurons demonstrated striking conservation with two neurons housed in each capitate peg sensillum in both male and female maxillary palp, aligning with previous findings in both *Ae. aegypti* and *An. coluzzii* [41,42].

### *Orco* knockout affects expression of receptors in *Ae. albopictus*

To investigate the impact of *Orco* mutations on tuning ORs expression in *Ae. albopictus*, we conducted transcriptomic analysis by antennal RNA-seq to compare expression of ORs tuning receptors in female wild-type and *AalbOrco*$^{DsRed/DsRed}$ homozygous mutants. As described in this study, as the T2A-QF2-3xP3-DsRed element was inserted in an exon of *Orco*,

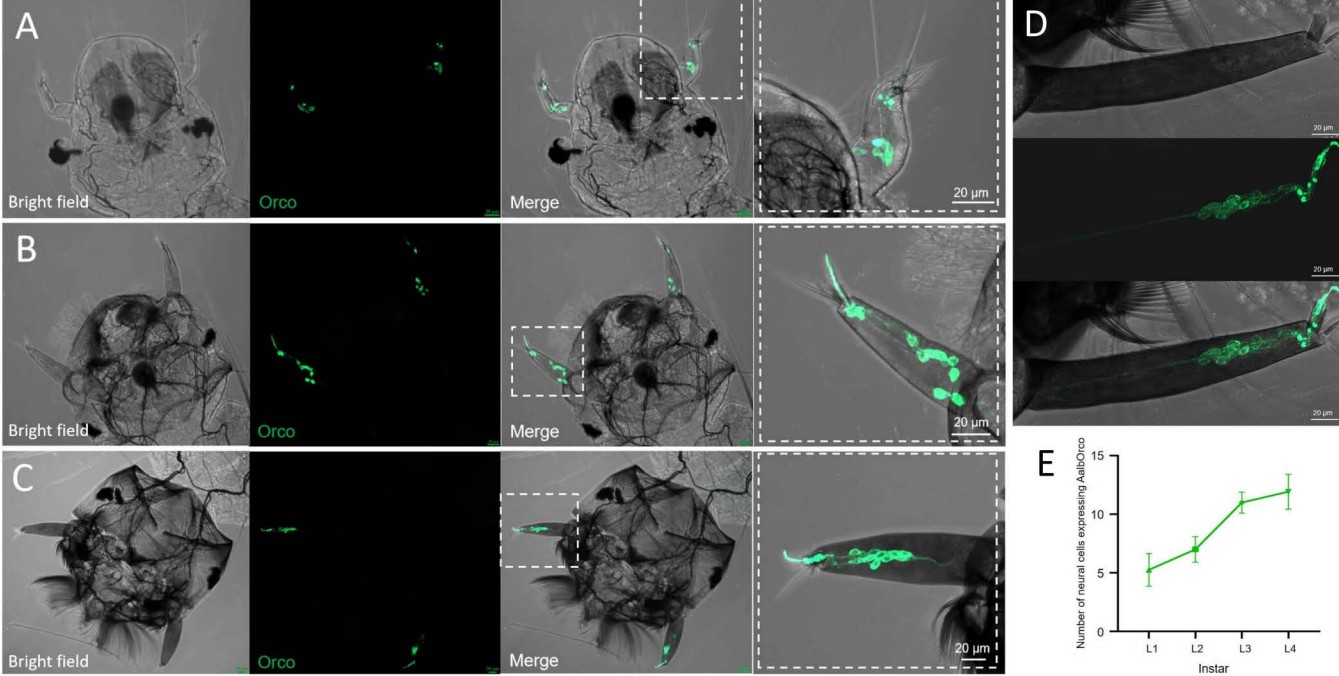

**Fig 3. Localization of *Orco*-expressing neurons in the larval antenna of *Ae. albopictus*. (A - D)** Maximum intensity projections of z-stack images showing GFP-labeled neurons in the antennae of 1-4 instar larvae. Scale bar: 20 μm. **(E)** Quantitative analysis of *Orco*⁺ neurons in 1-4 instar larvae. The number of *Orco*⁺ neurons increases with developmental stage, $N_{L_1} = 5.25 \pm 1.40$ (n = 8); $N_{L_2} = 7 \pm 1.10$ (n = 6); $N_{L_3} = 11 \pm 0.89$ (n = 6); $N_{L_4} = 11.93 \pm 1.49$ (n = 14).

which directly disrupted its normal function, we self-crossed the heterozygous *AalbOrco⁺/DsRed Ae. albopictus* and then obtained the homozygous *AalbOrco^DsRed/DsRed* mosquito based on the strength of fluoresence in the mosquito eye which was further confirmed by the sequencing results of genotyping. While a moderate reduction in egg laying and embryo hatching rate of individual female *AalbOrco^DsRed/DsRed* mosquito was observed (S1 Fig; S2 Table), we encountered no problem in maintaining the population of *AalbOrco^DsRed/DsRed* mosquito under laboratory condition.

Antennal RNA extracted from the wild-type and *AalbOrco^DsRed/DsRed* mosquitoes four day post eclosion was submitted for next-generation transcriptional profiling, and receptor transcript abundance was quantified with software FeatureCounts (S3 Table, "FPKM"). As expected, *Orco* expression is significantly reduced in *AalbOrco^DsRed/DsRed* mutants mosquitoes (Fig 5A, B). We found that the knockout of *Orco* leads to significantly reduced expression of many ORs tuning receptors (Fig 6A). The extremely modest number of residual *Orco* transcripts are most likely derived from the sequence in front of the double stranded break located in the fourth exon of the mutant mosquitos (Fig 5B). More importantly, the greatest reduction is seen for OR52, whose transcript in *AalbOrco^DsRed/DsRed* mutants is almost undetectable (Fig 5C). The transcript abundance of many other tuning receptors was also significantly diminished, with less than 20% of wild-type level expression, including: OR7a (1%), OR13a (5%), OR132 (6%), OR56a (8%), OR43 (8%), OR6 (8%), OR10 (17%), OR122 (17%), OR84 (18%) and OR111 (19%) (Fig 5C). In contrast, no significant differences were observed in the transcript abundance of IRs in the antennae of *AalbOrco^DsRed/DsRed* mosquito (Fig 5D). Interestingly, 2 *Or* transcripts, OR115 and OR85, exhibited upregulation, with significant increases of 87% and 60%, respectively, relative to wild-type controls (Fig 5F).

Previous studies in *Drosophila* and mosquitoes have identified the expression of certain GRs in the antenna, such as the $CO_2$-sensitive GRs in the ab1 sensillum of fly antenna [43] and GR33 in mosquito antenna [44]. In our

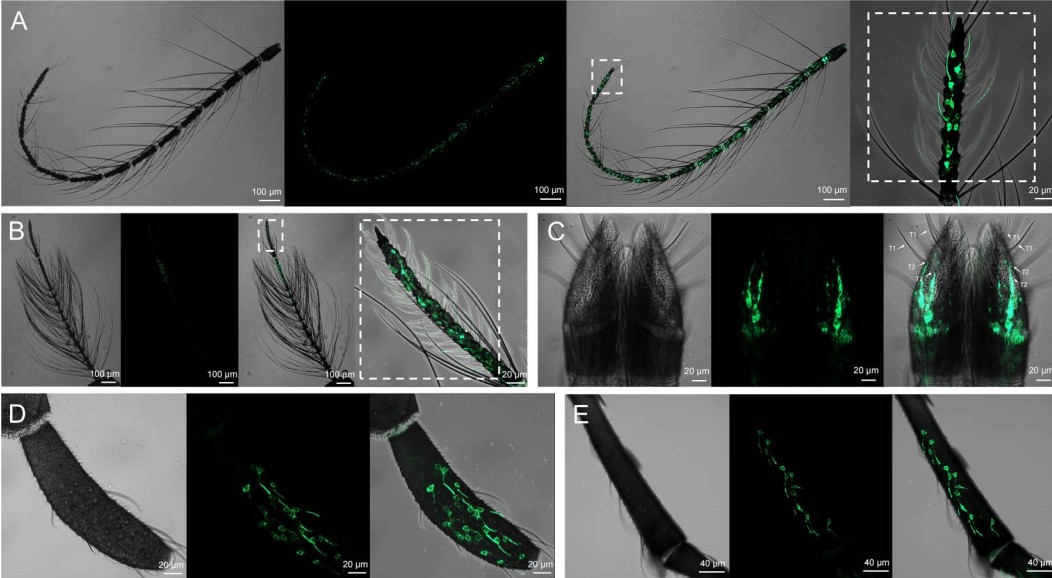

**Fig 4. Localization of *Orco* in olfactory appendages of *Ae. albopictus*. (A and B)** Representative confocal z-stack of *Orco*-expressing neurons in the antenae of female (A) and male (B) mosquitoes which exhibits sexual dimorphism with *Orco* expressed in all 13 segments of the female antennae while only in the distal two segments of the male antennae. **(C)** A representative confocal z-stack of a whole-mount female labellum showing *Orco* is expressed in T2 sensilla but not T1 sensillum (highlighted by arrows; scale bars, 10 μm). **(D and E)** Representative confocal z-stack of *Orco*-expressing neurons in the capitate peg sensilla of maxillary palp of both female (D) and male (E) mosquito.

study, we detected five GR transcripts of exceptionally low abundance in the antenna of both *Orco* homozygous mutants and wild-type mosquitoes, although these notably exhibited no significant difference between the two mosquito lines (Fig 5E).

### Reduced antennal responses of *Orco* knockout mosquito to human volatiles

To investigate the impact of *Orco* knockout on the compound-evoked olfactory responses of *Ae. albopictus*, we performed comparative electroantennogram (EAG) recordings between wild-type and *AalbOrco^DsRed/DsRed* homozygous mutant mosquitoes with a chemically diverse odorant panel (50 compounds across 7 chemical categories) which were selected according to their potent electrophysiological effects in other mosquito species [25,45,46] (S1 Table). As expected, wild-type mosquitoes exhibited robust EAG responses across all of these chemical groups (Fig 6A). In contrast, EAG responses of *AalbOrco* mutant mosquito to most odorants, particularly those in the categories of alcohol, aldehydes, ketones, and esters, were dramatically reduced (Fig 6B; S4 Table). Of note, we also observed significantly reduced EAG responses to some compounds in the heterozygous *AalbOrco^+/DsRed* mutants compared to the wild-type, which may result from the down-regulation of certain ORs induced by *Orco* knockout (S2 Fig). As expected, no significant differences were observed between *AalbOrco^DsRed/DsRed* and wild-type in response to three acids, valeric acid, hexanoic acid, and butanoic acid, which are often considered to be detected by IRs, suggesting the *Orco/OR* complex is not involved in the perception of these acidic compounds. These results provide strong evidence that *Orco* is necessary for olfactory chemosensation in *Ae. albopictus*.

### Reduced neuronal responses in *Orco* mutant *Ae. albopictus*

Four morphological types of trichoid sensilla have been previously identified on the antennae of *Ae. albopictus* mosquitoes, namely long sharp tipped (LST), short sharp tipped (SST), short blunt tipped I (SBTI), and short blunt tipped II

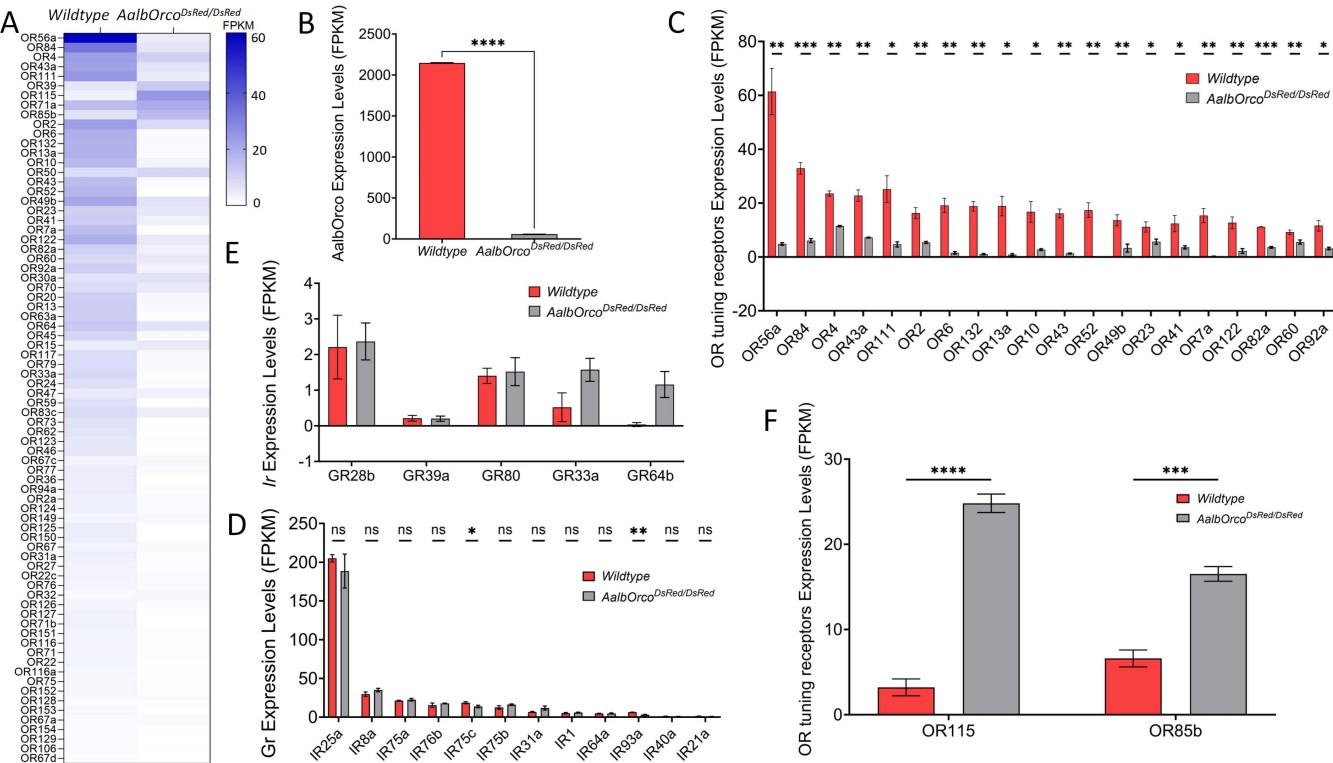

**Fig 5. Comparison of expression of *Or* tuning receptors between the wild-type and *AalbOrco*^DsRed/DsRed homozygous mutant mosquitoes.** Heatmap showing the average antennal expression in FPKM for each of the *Or* genes in wild-type and *AalbOrco*^DsRed/DsRed mosquitoes. (B) Transcriptional level of *Orco* gene in wild-type and *AalbOrco*^DsRed/DsRed mosquitoes. (C) Top 20 most highly expressed *Or* genes in wild-type were significantly reduced in *AalbOrco*^DsRed/DsRed mosquitoes. (D) Expression in FPKM for detected *IR* tuning receptor and co-receptor genes in wild-type and *AalbOrco*^DsRed/DsRed mosquitoes. (E) Expression in FPKM for detected GR genes in wild-type and *AalbOrco*^DsRed/DsRed mosquitoes. (F) Expression of all the two upregulated *Or* tuning receptors genes in the *AalbOrco*^DsRed/DsRed mosquitoes. Mann-Whitney U test was applied in the statistical analysis, statistical significance is presented as P<0.05 (*), P<0.01 (**), P<0.001 (***), P<0.0001(****) and P>0.05 (ns).

(SBTII) [47]. To investigate the effects of *Orco* knockout on chemical reception at the neuronal level, we performed SSR to compare the responses of four types of olfactory sensilla in wild-type and *Orco* mutants (*AalbOrco*^DsRed/DsRed) to a panel of 34 compounds in diverse chemical categories. We found in female wild-type *Ae. albopictus*, sensilla SST and SBTI exhibit broad neuronal responses to most compounds while LST and SBTII sensilla only respond to a limited number of chemicals (Fig 7A; S5 Table). However, in the *AalbOrco*^DsRed/DsRed mosquito, the trichoid sensilla, with the exception of SBTII, all exhibited loss of responsiveness to most compounds (Fig 7B; S5 Table).

The 'A' neuron, as defined by its large spike amplitude, in SST (blue traces in Fig 7C) displayed strong responses to 2,6-dimethylpyrazine (124±54 spikes/s), 4,5-dimethylthiazole (120±72 spikes/s), acetophenone (88±44 spikes/s), and 1-pentanol (82±62 spikes/s). Meanwhile, SBTI was very sensitive to terpenoid volatiles fenchone (82±20 spikes/s) and camphor (72±26 spikes/s) (Fig 7C, left panel). Moreover, the background activity of sensory neurons in the sharp-tiped sensilla (SST and LST) were considerably diminished with very few residual spikes present (Fig 7C, right panel). However, we did observe extensive spontaneous activity in the SBTI of the *Orco* mutant mosquito. Most interestingly, the SBTII sensilla are found to respond robustly to aldehydes, such as octanal, nonanal and decanal, in both wild-type and *Orco* mutant. The simple interpretation is that *Orco* is not involved in the sensation of aldehyde compounds in the SBTII sensillum of *Ae. albopictus*. Quantitative analysis on the neuronal response of *Orco* mutant mosquito revealed significant

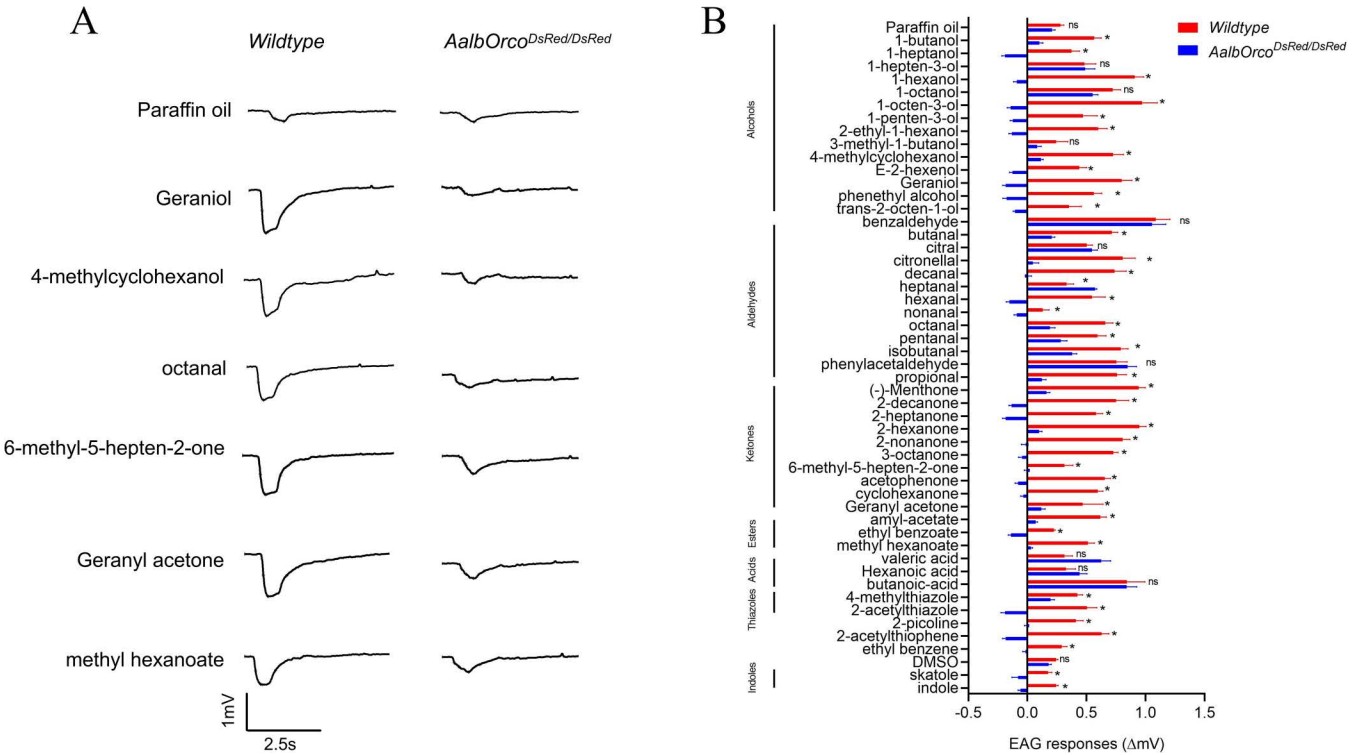

**Fig 6. EAG responses of wild-type and *AalbOrco*^DsRed/DsRed *Ae. albopictus* to a broad panel of human odorants. (A)** Representative EAG response traces of wild-type and *AalbOrco*^DsRed/DsRed mosquito to different odorants; **(B)** Comparison of EAG responses of wild-type and *AalbOrco*^DsRed/DsRed *Ae. albopictus* to 50 odorants in different chemical classes (n = 7). EAG responses (ΔmV) for each odorant at a $10^{-1}$ dilution were normalized to the solvent control (Paraffin oil and DMSO were used as solvents. Indole and skatole were dissolved in DMSO, while the other 48 compounds were dissolved in Paraffin oil) by subtracting the solvent-induced EAG value. Mann-Whitney U test was applied in the statistical analysis, with P ≥ 0.05 indicating no significance (ns), and P < 0.05 (*) as significant difference.

attenuation in SST sensitivity to 2,6-dimethylpyrazine, 4,5-dimethylthiazole, acetophenone, and 1-pentanol (Fig 7D). In addition, SBTI also exhibited significantly diminished responses to fenchone and camphor (Fig 7E). These results demonstrate that *Orco-expressing* neurons housed in most trichoid sensilla are actively involved in the detection of a wide range of compounds.

Furthermore, we examined the odor-evoked responses of capitate peg (CP) sensilla which house three neurons, A neurons showing large spikes and B/C neurons displaying small spikes, on the maxillary palps of both wild-type and *Orco* mutant *Ae. albopictus*. The GR-expressing A neurons are responsible for detecting $CO_2$ while the B/C neurons expressing the Orco/OR complex are extremly sensitive to one human sweat component: 1-octen-3-ol [24,48,49]. Our results revealed that in wild-type mosquitoes, the B/C neurons in CP sensilla of *Ae. albopictus* exhibited significantly higher responses to 1-octen-3-ol (47 ± 10 spikes/s) compared to *Orco* mutants, where the response to 1-octen-3-ol was almost completely abolished (Fig 7F). In contrast, the $CO_2$-detecting A neuron remained unaffected with both wild-type and *Orco* mutant mosquito displaying robust $CO_2$-induced responses (Fig 7F).

## Impact of *Orco* knockout on the behavioral responses of mosquitoes

To further investigate the impact of *AalbOrco*^DsRed/DsRed mutation on blood-feeding and host preference, we conducted blood-feeding assays using mice as the blood source. We observed a significant decline in the proportion of successful

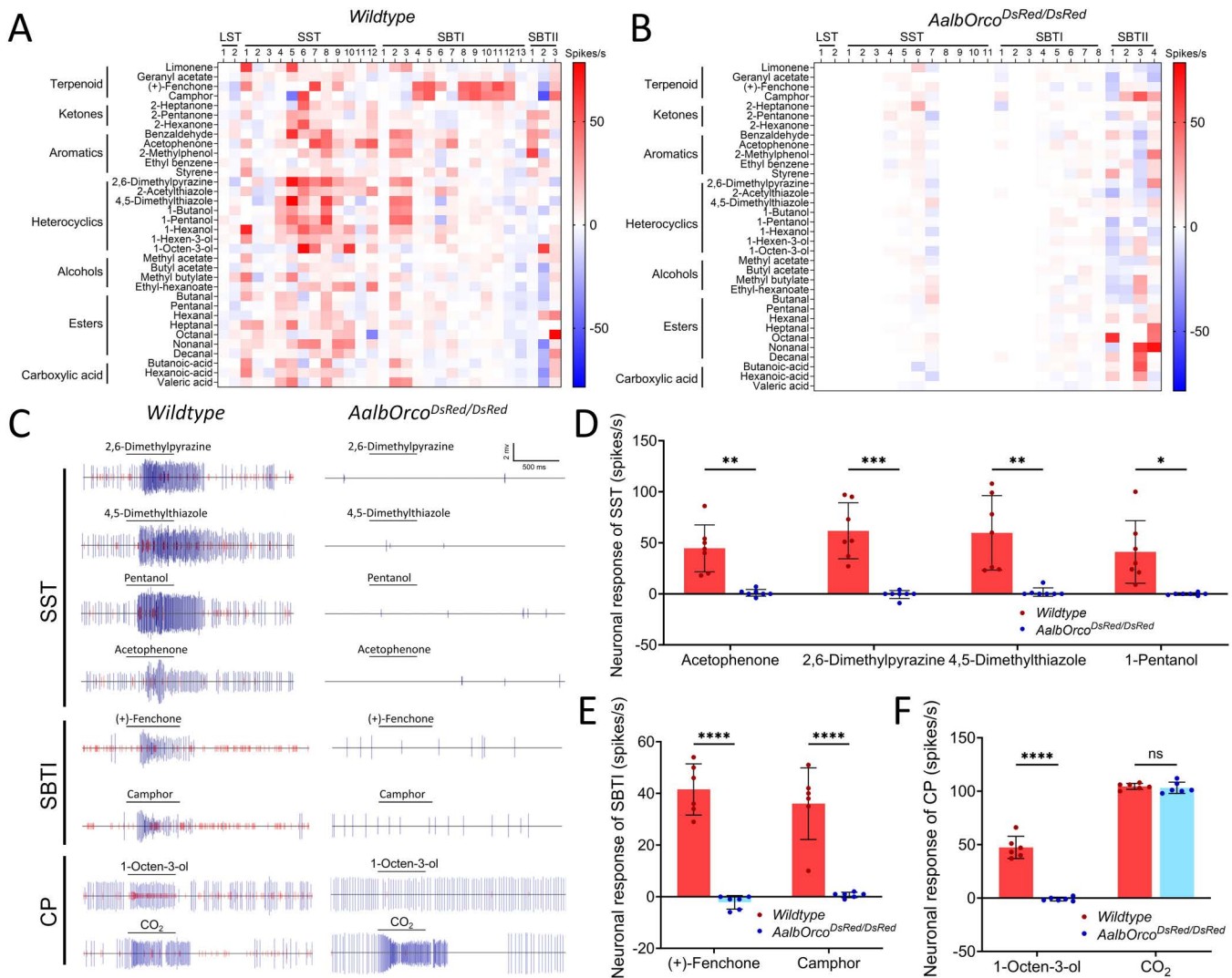

**Fig 7. Responses of ORNs of different trichoid sensillum of wild-type and *Orco* mutant mosquito to odorants. (A)** & **(B)** Heatmaps showing averaged neuronal responses to a panel of 34 odorants from individual sensilla randomly sampled across the antenna of wild-type (left) and *AalbOrco^{DsRed/DsRed}* (right) *Ae. albopictus*. **(C)** Representative response traces of SST and SBTI sensillum to selected chemical compounds (diluted to 1% concentration in paraffin oil or DMSO). There were two neurons in most of the sensilla with large amplitude (blue traces) representing the response of 'A' neuron and small amplitude (red traces) representing the response of 'B' neuron. **(D)** Averaged female SST sensillum responses in wild-type and *AalbOrco^{DsRed/DsRed}* mosquitoes. Mann-Whitney U test suggested neuronal responses to selected compounds in *Orco* mutants were significantly reduced compared to the wild-type (n = 7). **(E)** Averaged female SBTI sensillum responses in wild-type and *AalbOrco^{DsRed/DsRed}* mosquitoes. Mann-Whitney U test suggested neuronal responses to fenchone and camphor in *Orco* mutants were significantly reduced than wild-type (n = 6). **(F)** Averaged female CP sensillum responses in the the maxillary palp of wild-type and *AalbOrco^{DsRed/DsRed}* mosquitoes. Mann-Whitney U test suggested neuronal responses to 1-octen-3-ol in *Orco* mutants were significantly reduced compared to the wild-type (n = 6), while no significant differences were observed in neuronal responses to $CO_2$ between wild-type and *Orco* mutants (n = 6). Statistical significance was presented as $P < 0.05$ (*), $P < 0.01$ (**), $P < 0.001$ (***), $P < 0.0001$ (****).

blood-feeding in *AalbOrco^{DsRed/DsRed}* mutants compared to the wild-type mosquitoes ($P < 0.05$) (Fig 8A, B; S6 Table). In addition, a two-choice valence assay between volatile odors emitted from a human hand and that of a mouse was utilized to evaluate host preference of both wild-type and *Orco* knockout mosquitoes. As expected, wild-type mosquitoes displayed considerable preference towards the human hand. By contrast, host preference towards humans decreased in

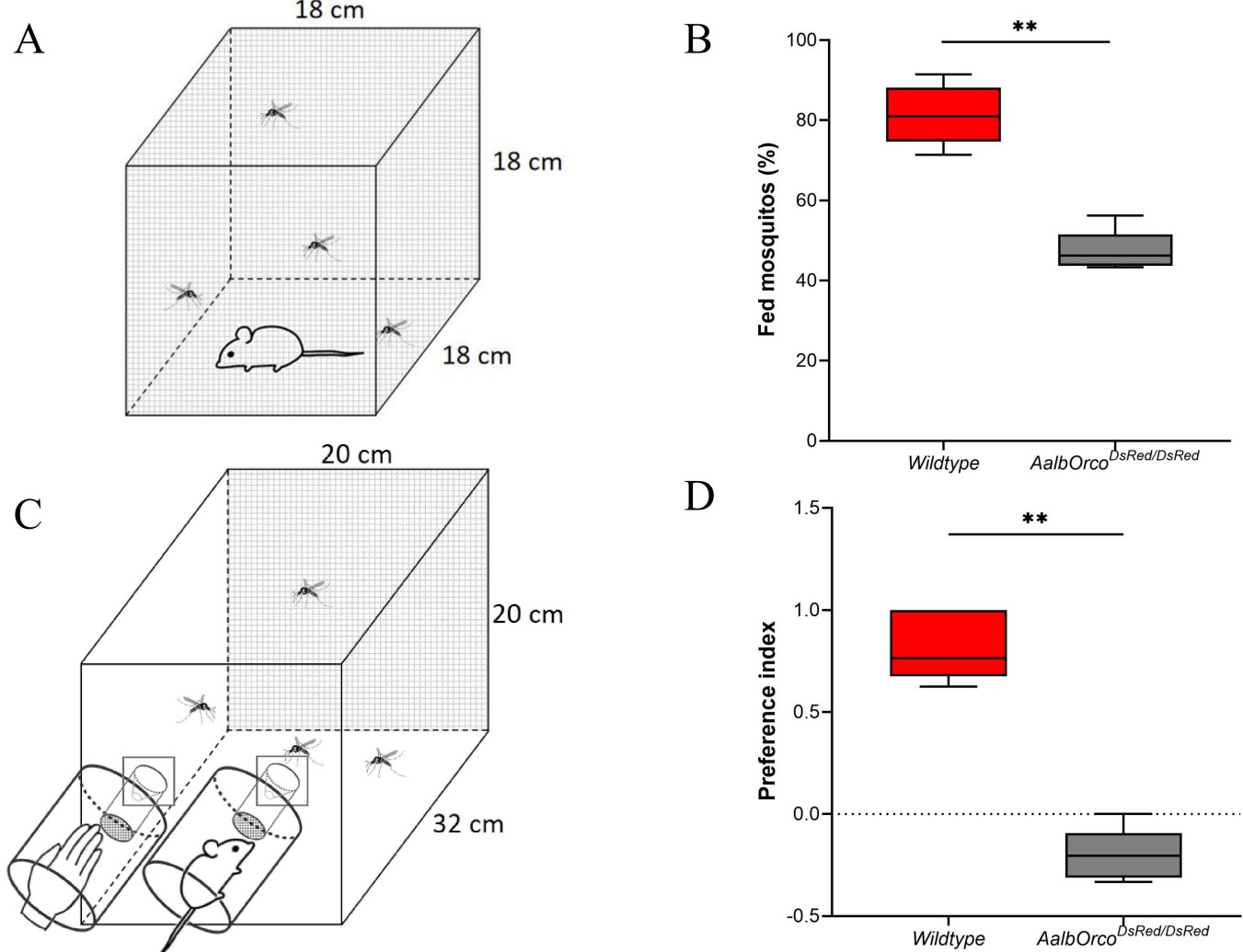

**Fig 8. *Orco* knockout-induced changes in the mosquito blood-feeding and host-seeking behaviors. (A)** Schematic picture of setting of blood-feeding assay. **(B)** Comparison of feeding success between *AalbOrco*$^{DsRed/DsRed}$ and wild-type mosquitoes. The successful blood-feeding rate of the *AalbOrco*$^{DsRed/DsRed}$ mosquito was significantly lower than that of wild-type mosquitoes on mice (P<0.05). **(C)** Schematic picture of setting of two-choice host preference assay. **(D)** Comparison of host preference between *AalbOrco*$^{DsRed/DsRed}$ and wild-type mosquitoes. *AalbOrco*$^{DsRed/DsRed}$ mutant mosquitoes showed reduced preference for human hand (P<0.0001). Statistical analysis was done using GraphPad Prism 5. Data are presented as mean±s.e.m. Mann-Whitney U test was used to compare two sets of data with the significance set to a P-value<0.05.

*AalbOrco*$^{DsRed/DsRed}$ mutant mosquitoes (P<0.0001) which instead showed a slight preference to the mice in our study (Fig 8C, D; S6 Table). Taken together, these findings make *Orco* a prime target for vector control strategies aimed at disrupting blood feeding and host preference in *Ae. albopictus*.

## Discussion

The emergence and global spread of the Asian tiger mosquito *Ae. albopictus* as a dominant arboviral vector represents a growing threat to public health, particularly given its expanding ecological range and capacity to transmit deadly diseases [1,2,8,50]. Despite the critical role of olfaction in the mosquito's ability to locate hosts and reproduce, fundamental

questions remain about how the olfactory system develops and functions in this species. In this study, we provide a systematic and comprehensive analysis of *Orco* expression, function, and regulatory impact across the life stages of *Ae. albopictus*, revealing an unexpectedly dynamic olfactory architecture and underscoring *Orco*'s multifaceted role in olfactory development, transcriptional regulation, and behavior.

Our study establishes that *Orco*-expressing OSNs are present as early as the embryonic stage and expand throughout larval development, suggesting an early onset of olfactory capability that may contribute to environmental sensing, egg hatching, and larval foraging [40]. Here, we observed an average of 11 *Orco*⁺ neurons in the third and fourth instar larval antenna, which is consistent with what has been reported in other mosquito species. For instance, the sensory cone in the larval antennae of *Ae. aegypti*, *An. coluzzii*, and *An. gambiae* (previously known as the "M" and "S" forms of the *An. gambiae* species complex) [51] have all been reported to innervate 12–13 typical bipolar neurons that express Orco/OR complex and display robust chemosensory capacity [52–55]. Compared to several hundred Orco+ neurons in the antenna of adult mosquitoes, the modest number of ORNs in the larval antenna may mirror the relatively less complex chemical environment of their aquatic habitat—characterized by a limited diversity of volatile chemical cues and a primary reliance on water-soluble or non-volatile compounds—along with the correspondingly narrower behavioral repertoire required for larval survival (e.g., foraging, habitat navigation, and predator avoidance).

Functionally, *Orco* mutants exhibited a profound deficit in olfactory physiology. EAG and SSR demonstrated sharply reduced responses across multiple classes of odorants, including key human-derived volatiles such as 1-octen-3-ol and other aldehydes and ketones [13]. Indeed, the neuronal responses of most of the trichoid sensilla investigated in this study were almost completely abolished after *Orco* knockout. These deficits were sensory-specific, with GR- and IR-mediated responses largely preserved, reinforcing the selectivity of the *Orco*/*OR* pathway and corroborating earlier findings in *Anopheles* and *Aedes* species [24,25]. Behavioral assays further confirmed that *Orco* disruption abolishes host preference and significantly reduces blood-feeding success, firmly establishing *Orco* as a central node linking molecular, cellular, and behavioral levels of mosquito olfaction.

Curiously, we also observed abundant residual spikes in the sensilla of *Orco* mutants, particularly in the SBTI sensilla, which displayed a very high frequency of spontaneous sensory neuron activity. Our best explanation for these residual spikes comes from recent studies reporting non-canonical co-expression of chemoreceptor co-receptors (e.g., *Orco*, *Ir25a*, *Ir8a* and *Ir76b*) and/or tuning receptors in the olfactory sensory neurons of both *Drosophila* and mosquitoes [9–11,56,57]. The extensive presence of residual spikes in the SBTI sensillum of *Orco* mutant mosquito may therefore be the result of the co-expression of both *Orco*/*OR* and *Irco*/*IR* in the same neurons, which could potentially still be partially functional even though the *Orco*/*OR* complex has been disrupted. Future studies on olfactory receptor organization at a single cell level in *Ae. albopictus* would give us a more definitive explanation of these residual spikes displayed after knocking out the *Orco* gene.

Beyond anatomical remodeling and physiology, our transcriptomic analysis revealed that *Orco* knockout in *Ae. albopictus* led to widespread downregulation and targeted upregulation of many tuning ORs, implicating *Orco* not only in OSN signaling but also in the maintenance of ORs gene expression. These findings align with evidence suggesting a dual role for *Orco* in the transcriptional regulation of tuning ORs in *D. melanogaster* and *H. armigera* [58,59]. Together with a more recent study in *Ae. aegypti* [60], our results demonstrate that this dual regulatory role of *Orco* is conserved in mosquitoes, extending the functional landscape of *Orco* beyond its canonical role as merely a co-receptor [14,15]. The reduction in ORs transcript levels suggests that *Orco* may influence receptor stability, trafficking, or transcriptional feedback, an area ripe for further investigation.

## Conclusions

Taken together, this work provides the most detailed spatiotemporal characterization of *Orco* expression and function to date in a non-model mosquito species and establishes a new platform for studying olfactory system development.

By integrating genetic tools (Q-system, HACK), transcriptomics, neurophysiology, and behavior, we demonstrate that *Orco* serves not only as a co-receptor but as a transcriptional stabilizer in *Ae. albopictus*. These findings provide a theoretical foundation for translational strategies aimed at disrupting mosquito host-seeking behaviors via genetic or chemical interference with *Orco* function. Given the expanding public health burden of *Ae. albopictus* and the increasing resistance to traditional insecticides, *Orco* and its associated pathways represent compelling targets for next-generation vector control.

## Supporting information

**S1 Fig. Comparison of the fecundity between the wild-type and *AalbOrco*[DsRed/DsRed].** (A) Egg laying of individual female wild-type and *AalbOrco*[DsRed/DsRed] mosquito (n = 8). The number of eggs laid by *AalbOrco*[DsRed/DsRed] was lower than that of the wild-type. (B) Hatching rate of eggs laid by wild-type and *AalbOrco*[DsRed/DsRed] mosquito (n = 8). The hatching rate of eggs laid by *AalbOrco*[DsRed/DsRed] mosquito was lower than that of the wild-type. Mann-Whitney U test was applied in the statistical analysis, statistical significance is presented as $P < 0.05$ (*), $P < 0.01$ (**), $P < 0.001$ (***), and $P > 0.05$ (ns).
(TIF)

**S2 Fig. EAG responses of wild-type and *AalbOrco*[+/DsRed] *Ae. albopictus* to a broad panel of human odorants.** Comparison of EAG responses of wild-type and *AalbOrco*[+/DsRed] *Ae. albopictus* to 50 odorants in different chemical classes (n = 8). EAG responses (ΔmV) for each odorant at a $10^{-1}$ dilution were normalized to the solvent control (Paraffin oil and DMSO were used as solvents. Indole and skatole were dissolved in DMSO, while the other 48 compounds were dissolved in Paraffin oil.) by subtracting the solvent-induced EAG value. Mann-Whitney U test was applied in the statistical analysis, with $P \geq 0.05$ indicating no significance (ns), and $P < 0.05$ (*) as significant differences.
(TIF)

**S1 Table. Odorants lists used in electrophysiological recordings.** Chemical information of odorants, such as CAS number, purity, and company, was included.
(PDF)

**S2 Table. Source data for fecundity assay of both wild-type and *AalbOrco*[DsRed/DsRed] female mosquito.**
(XLSX)

**S3 Table. Source data for the transcript abundance (FPKM counts) of chemosensory genes in both wild-type and *AalbOrco*[DsRed/DsRed] female mosquito antennae.**
(XLSX)

**S4 Table. Source data for EAG responses on both wild-type and *AalbOrco*[DsRed/DsRed] female mosquito antennae.**
(XLSX)

**S5 Table. Source data for SSR recordings on both wild-type and *AalbOrco*[DsRed/DsRed] female mosquito antennae.**
(XLSX)

**S6 Table. Source data for behavioral assays on both wild-type and *AalbOrco*[DsRed/DsRed] female mosquito antennae.**
(XLSX)

## Acknowledgments

We thank the Biosafety Level 2 (BSL-2) and Animal Biosafety Level 2 (ABSL-2) Laboratories of the Institute of Infectious Diseases, Shenzhen Bay Laboratory, for their support and assistance during this research.

## Author contributions

**Conceptualization:** Zi Ye, Feng Liu.

**Formal analysis:** Hui Yao, Qian Qi, Dan Gou.

**Funding acquisition:** Qian Qi, Stephen T. Ferguson, Zi Ye, Feng Liu.

**Investigation:** Hui Yao, Qian Qi, Dan Gou.

**Resources:** Simin Liang.

**Writing – original draft:** Hui Yao, Qian Qi.

**Writing – review & editing:** Stephen T. Ferguson, Ming Li, Heng Zhang, Zi Ye, Feng Liu.

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
