## [Decision Letter · Decision Letter 0]

21 Jul 2025

Response to Reviewers
Revised Manuscript with Track Changes
Manuscript

Shaden Kamhawi

co-Editor-in-Chief

Paul Brindley

co-Editor-in-Chief

**Journal Requirements:**

At this stage, the following Authors/Authors require contributions: Hui Yao, Qian Qi, Dan Gou, Simin Liang, Stephen Ferguson, Heng Zhang, Zi Ye, and Feng Liu. Please ensure that the full contributions of each author are acknowledged in the "Add/Edit/Remove Authors" section of our submission form.

3) Your manuscript is missing the following sections: Results.  Please ensure all required sections are present and in the correct order. Make sure section heading levels are clearly indicated in the manuscript text, and limit sub-sections to 3 heading levels. An outline of the required sections can be consulted in our submission guidelines here:

5) We have noticed that you have cited Table  Table S2 in the manuscript file but there is no corresponding table in the manuscript.  Please amend your manuscript to include this table noting that tables should not be uploaded as individual files.

6) We have noticed that you have uploaded Supporting Information files, but you have not included a list of legends. Please add a full list of legends for your Supporting Information files after the references list.

7) Some material included in your submission may be copyrighted. According to PLOSu2019s copyright policy, authors who use figures or other material (e.g., graphics, clipart, maps) from another author or copyright holder must demonstrate or obtain permission to publish this material under the Creative Commons Attribution 4.0 International (CC BY 4.0) License used by PLOS journals. Please closely review the details of PLOSu2019s copyright requirements here: PLOS Licenses and Copyright. If you need to request permissions from a copyright holder, you may use PLOS's Copyright Content Permission form.

Potential Copyright Issues:

i) Please confirm (a) that you are the photographer of 1E, 1F, 2, 3, and 4, or (b) provide written permission from the photographer to publish the photo(s) under our CC BY 4.0 license.

ii) Figures 8A, and 8C. Please confirm whether you drew the images / clip-art within the figure panels by hand. If you did not draw the images, please provide (a) a link to the source of the images or icons and their license / terms of use; or (b) written permission from the copyright holder to publish the images or icons under our CC BY 4.0 license. Alternatively, you may replace the images with open source alternatives. See these open source resources you may use to replace images / clip-art:

8) In the online submission form, you indicated that All materials used in this study are available upon request from the corresponding author.. All PLOS journals now require all data underlying the findings described in their manuscript to be freely available to other researchers, either

1. In a public repository

2. Within the manuscript itself

3. Uploaded as supplementary information.

9) Kindly revise your competing statement to align with the journal's style guidelines: 'The authors declare that there are no competing interests.'

**Reviewers' comments:**

**Key Review Criteria Required for Acceptance?**

**Methods**

-Are the objectives of the study clearly articulated with a clear testable hypothesis stated?

-Is the study design appropriate to address the stated objectives?

-Is the population clearly described and appropriate for the hypothesis being tested?

-Is the sample size sufficient to ensure adequate power to address the hypothesis being tested?

-Were correct statistical analysis used to support conclusions?

-Are there concerns about ethical or regulatory requirements being met?

Reviewer #1: Objective need to be added clearly and reduce introduction

Reviewer #2: The methods are clear, detailed and sound.

Reviewer #3: Would there be an effect on the experiment when using sucrose water rather than the WHO recommended glucose water for feeding adult mosquitos?

Was there ethical clearance for blood feeding assay and host preference assay.

Which blood meal source was used to rear the Aedes colony at the Shenzhen Bay Laboratory Insectary?

**Results**

-Does the analysis presented match the analysis plan?

-Are the results clearly and completely presented?

-Are the figures (Tables, Images) of sufficient quality for clarity?

Reviewer #1: The figures are good quality

Reviewer #2: Yes

Reviewer #3: Well presented

**Conclusions**

-Are the conclusions supported by the data presented?

-Are the limitations of analysis clearly described?

-Do the authors discuss how these data can be helpful to advance our understanding of the topic under study?

-Is public health relevance addressed?

Reviewer #1: see for some adjusted in my detail comments

Reviewer #2: Yes, the conclusions made are supported by the data presented.

The limitations of the study are missing.

Reviewer #3: Well supported conclusions.

**Editorial and Data Presentation Modifications?**

Reviewer #1: (No Response)

Reviewer #2: The manuscript submitted by Yao and co-authors unveils the Odorant co-receptor (Orco) cellular localization patterns, developmental expression dynamics and impact on olfactory behavior of the Aedes albopictus mosquito. Authors have demonstrated the importance of Orco in the olfactory architecture of Ae. albopictus and its role in host-seeking behavior, which is a great milestone. Overall, the manuscript is well written and the data generated here is relevant to the vector research community. However, minor changes can be made before acceptance for publication.

Major comments

In the methods, I miss the rationale of using mice instead of a human subject for the blood-feeding assay.

Page 3 line 75-78: Rephase

Page 9 line 360-361: I also miss the rationale of having different starvation times for the blood-feeding (12hrs) and host preference assay (24hrs ) in page 10 line 377.

Page 9 line 342-343: Six libraries were generated. Are the transcriptomes available to the public?

Page 12 line 454: “AalbOrco expression in the labella was exclusively localized to ORNs with…” Do you mean for females or males?

Page 14 line 561: I believe it is not accurate to state that “We observed a significant decline in blood-feeding efficiency in AalbOrcoDsRed/DsRed mutants compared to the wildtype mosquitoes”. While the p-value may indicate significance, the double mutants have a feeding efficiency of about 50%, which is more than half of the wild type. There is a relative reduction in feeding efficiency in the double mutants, but these results might also differ if the feeding assay is repeated with human subjects, who are the preferred hosts. Then it would be more relevant from a public health perspective. I believe the data shows feeding success rather than efficiency. A box plot might also present the data better.

Figure 3: The scales have the same numerical value but different lengths.

Figure 6A: Include the results for the diluent.

In the results and figures indicate whether its female/male or mutant/wildtype mosquito tha were used for results presented.

Were there any notable changes in general behavior and probing frequency in the mutants as compared to the controls?

Minor comments

Page 5 line 153: I guess you mean lesser-studied species instead of lesser-studies.

Page 5 line 181: Can read “7-day-old non-blood-fed females” instead of “7-day-old without blood meal fed..”

Page 7 line 272: Do you mean mutant females or wildtype?

Page 9 line 343: “three repeats…..” should read “three replicates”

Page 9 line 353: “FPKM” to “FPKM counts”

Reviewer #3: (No Response)

**Summary and General Comments**

Reviewer #1: Hui Yao et al, The manuscript entitled: Unveiling the Developmental Dynamics and Functional Role of Odorant Receptor Coreceptor (Orco) in Aedes albopictus: A Novel Mechanism for Regulating Tuning Odorant Receptor Expression

The authors described and elucidated the expression and role of Orco in Aedes albopictus olfaction and showed its tissue and developmental expression they also showed reduced expression of most Ors when Orco is knockdown, but the Irs and Grs, which are independent of Orco remains unaffected. Similarly, response of Ors significantly reduced but Irs response to tested acids remain unaffected, which is expected as Orco is conserved both in its expression and function.

Q1. However, there are interesting results two Or115 and Or85 over expressed as compared to wildtype in SBTII and their response is significantly high, the speculation is not convincing as Orco role is highly conserved, may be unintended consequence, artifact specifically in that sensilla, may be give a better speculation.

Q2. EAG: how inhibition happens in EAG Fig 6B, is that normalized EAG as response of treatment-solvent??

Q3. Fig 8D: Preference index is not clear, feeding(Fig 8B) is fine even though it is reduced, the figure shows preference shitted from human to mouse, I was expecting undifferentiated response to both sides, but now the mutant responded to mouse than human , that is why the response index is negative

Q4. Did you try how is the response of either physiology or behaviour of heterozygous individuals?

Q5. What is the control for the mutation like parental control?

Q6. Any adverse effect on lifespan, reproduction of the Orco mutation?

Q7. Line 66-68 ........efficiency, and elimination of host preference in

females, there is no elimination they are feeding 50% of them fed( Fig 8B) but host preference shift is observed but no elimination

Q8. Line 491: While GRs are generally considered absent from antennal expression. Correction: Grs are expressed in a given sensilla found on antenna, need to be rewritten

Minor

Introduction: too long introduction summarize the story by mentioning the why of the experiment

Terminology: Tuning Odorant Receptor Expression, the word tuning should be deleted

Reviewer #2: (No Response)

Reviewer #3: This study delivers a comprehensive spatiotemporal map of Orco expression and function in a the mosquito, revealing its dual role as an olfactory co-receptor and transcriptional stabilizer in Aedes albopictus and offering new avenues for disrupting host-seeking behavior.

Line 593 "the relatively less complex chemical environment of the aquatic system" This statement would require expounding on the 'less complex'.

PLOS authors have the option to publish the peer review history of their article (what does this mean? ). If published, this will include your full peer review and any attached files.

**Do you want your identity to be public for this peer review?** For information about this choice, including consent withdrawal, please see our Privacy Policy .

Reviewer #1: No

Reviewer #2: No

Reviewer #3: No

**Figure resubmission:****Reproducibility:** To enhance the reproducibility of your results, we recommend that authors of applicable studies deposit laboratory protocols in protocols.io, where a protocol can be assigned its own identifier (DOI) such that it can be cited independently in the future. Additionally, PLOS ONE offers an option to publish peer-reviewed clinical study protocols. Read more information on sharing protocols at https://plos.org/protocols?utm_medium=editorial-email&utm_source=authorletters&utm_campaign=protocols

---

## [Decision Letter · Decision Letter 1]

9 Nov 2025

Dear Dr. Liu,

We are pleased to inform you that your manuscript 'Unveiling the Developmental Dynamics and Functional Role of Odorant Receptor Co-receptor (Orco) in Aedes albopictus: A Novel Mechanism for Regulating Tuning Odorant Receptor Expression' has been provisionally accepted for publication in PLOS Neglected Tropical Diseases.

Best regards,

Paul O. Mireji, PhD

Section Editor

Paul Mireji

Section Editor

Shaden Kamhawi

co-Editor-in-Chief

Paul Brindley

co-Editor-in-Chief

Reviewer's Responses to Questions

**Key Review Criteria Required for Acceptance?**

**Methods**

-Are the objectives of the study clearly articulated with a clear testable hypothesis stated?

-Is the study design appropriate to address the stated objectives?

-Is the population clearly described and appropriate for the hypothesis being tested?

-Is the sample size sufficient to ensure adequate power to address the hypothesis being tested?

-Were correct statistical analysis used to support conclusions?

-Are there concerns about ethical or regulatory requirements being met?

Reviewer #1: (No Response)

Reviewer #2: Yes

Reviewer #3: (No Response)

**Results**

-Does the analysis presented match the analysis plan?

-Are the results clearly and completely presented?

-Are the figures (Tables, Images) of sufficient quality for clarity?

Reviewer #1: (No Response)

Reviewer #2: Yes

Reviewer #3: (No Response)

**Conclusions**

-Are the conclusions supported by the data presented?

-Are the limitations of analysis clearly described?

-Do the authors discuss how these data can be helpful to advance our understanding of the topic under study?

-Is public health relevance addressed?

Reviewer #1: (No Response)

Reviewer #2: Yes

Reviewer #3: (No Response)

**Editorial and Data Presentation Modifications?**

Reviewer #1: (No Response)

Reviewer #2: None

Reviewer #3: (No Response)

**Summary and General Comments**

Reviewer #1: (No Response)

Reviewer #2: The article is well-organized, and all concerns raised were addressed.

Reviewer #3: (No Response)

PLOS authors have the option to publish the peer review history of their article (what does this mean? ). If published, this will include your full peer review and any attached files.

**Do you want your identity to be public for this peer review?** For information about this choice, including consent withdrawal, please see our Privacy Policy .

Reviewer #1: No

Reviewer #2: No

Reviewer #3: No

---

## [Editor Report · Acceptance letter]

21 Nov 2025

Dear Dr. Liu,

We are delighted to inform you that your manuscript, "Unveiling the Developmental Dynamics and Functional Role of Odorant Receptor Co-receptor (Orco) in Aedes albopictus: A Novel Mechanism for Regulating Tuning Odorant Receptor Expression," has been formally accepted for publication in PLOS Neglected Tropical Diseases.

Best regards,

Shaden Kamhawi

co-Editor-in-Chief

Paul Brindley

co-Editor-in-Chief

PLOS Neglected Tropical Disease